# Melanoma and Vitiligo: In Good Company

**DOI:** 10.3390/ijms20225731

**Published:** 2019-11-15

**Authors:** Cristina Maria Failla, Maria Luigia Carbone, Cristina Fortes, Gianluca Pagnanelli, Stefania D’Atri

**Affiliations:** 1Experimental Immunology Laboratory, IDI-IRCCS, 00167 Rome, Italy; 2Epidemiology Unit, IDI-IRCCS, 00167 Rome, Italy; 31st Dermatology Division, IDI-IRCCS, 00167 Rome, Italy; 4Molecular Oncology Laboratory, IDI-IRCCS, 00167 Rome, Italy

**Keywords:** melanoma, autoimmunity, prognostic markers

## Abstract

Cutaneous melanoma represents the most aggressive form of skin cancer, whereas vitiligo is an autoimmune disorder that leads to progressive destruction of skin melanocytes. However, vitiligo has been associated with cutaneous melanoma since the 1970s. Most of the antigens recognized by the immune system are expressed by both melanoma cells and normal melanocytes, explaining why the autoimmune response against melanocytes that led to vitiligo could be also present in melanoma patients. Leukoderma has been also observed as a side effect of melanoma immunotherapy and has always been associated with a favorable prognosis. In this review, we discuss several characteristics of the immune system responses shared by melanoma and vitiligo patients, as well as the significance of occurrence of leukoderma during immunotherapy, with special attention to check-point inhibitors.

## 1. Introduction

Melanoma is the most threatening form of skin tumor and its incidence is constantly increasing in the Western population. In Europe, more than 100,000 melanoma cases are registered each year [1]. General characteristics of cancer cells are genome instability and mutation load that should in principle render them good targets for the host immune system [2]. Among the tumor types, melanoma is considered a highly immunogenic tumor due to its elevated mutation load [3]. Accordingly, the presence of melanoma-specific cytotoxic T lymphocytes (CTLs) in the blood and in the skin that surround the tumor indicate that melanoma cells do not evade immune recognition. Moreover, the frequency of CTLs recognizing melanoma antigens appears to be higher in patients with metastatic disease than in those with primary tumors [4], suggesting that an increment in the antigen load is associated with tumor progression and can be recognized by the host immune system. Nevertheless, even when CTL responses occur, patients’ immune system is rarely in the condition of mounting an effective reaction, leading to tumor clearance [5]. It is now evident that tumor development, also in the case of melanoma, is accompanied by an impairment of the host immune system. In fact, immunodeficiency is associated with a higher incidence of melanocytic nevi and of melanoma [6], supporting the hypothesis of the existence of an active immunosurveillance against melanocytic proliferation in healthy individuals [7].

Tumor cells use different molecular strategies to elude host immune responses [8]. Cytokines secreted by tumor cells alter dendritic cell maturation and render them unable to elicit antigen-specific CTLs. Moreover, immature dendritic cells mediate tumor tolerance by inducing anergy of CTLs and expansion of T regulatory lymphocytes. Programmed death ligand 1 (PD-L1) is a major negative regulatory ligand that engages its programmed death 1 (PD-1) receptor expressed on activated T cells. PD-L1 can be expressed on immature tumor-associated dendritic cells, negatively affecting their functions. PD-L1 can be also expressed by tumor endothelial cells, producing an immunosuppressive environment, and by tumor cells themselves, blocking CTL activity and tumor rejection [9]. Cytokines and chemokines produced by tumor cells can inhibit T lymphocyte crossing of the tumor vasculature by reducing endothelial cell expression of adhesion molecules, while functioning as chemoattractants for immunosuppressive leukocytes such as T regulatory cells, Tie2+ monocytes, or myeloid-derived suppressor cells. In addition, hypoxic condition and acidosis that characterize the tumor microenvironment contribute in negatively affecting CTL functions [10].

Skin depigmentation, such as that occurring in vitiligo or Sutton’s nevus, has been associated with cutaneous melanoma since the 1970s [11]. Most of the antigens recognized by CTLs isolated from melanoma patients are expressed by both melanoma cells and normal melanocytes, explaining why autoimmune responses against melanocytes that lead to vitiligo or Sutton’s nevus could also be present in melanoma patients, resulting in melanoma-associated leukoderma or halo phenomena. Antigens recognized by CTLs include proteins of the melanogenic pathway, such as gp100, MART-1, tyrosinase, and tyrosinase-related proteins 1 and 2 [12,13,14]. Autoimmune responses against such antigens occasionally get into a specific immune reaction against melanoma and into tumor regression. Molecular identification of melanoma antigens recognized by CTLs represented the first step towards vaccination approaches against melanoma [15,16]. Notably, melanoma-associated leukoderma appeared as an adverse effect of these anti-melanoma therapies [17,18]. Together with therapeutic vaccination, additional trials were represented by the systemic treatment with interferons (IFN) or chemokines to stimulate the endogenous immune system to fight against the tumor. Interestingly, interleukin (IL)-2-based immunotherapy showed that a high percentage of responding patients developed skin depigmentation together with melanoma regression [19]. More recently, Curti et al. reported data about immune-related adverse events (irAEs) for high dose IL-2 therapy from the PROCLAIM registry [20]. They found that irAE following IL-2 therapy was associated with tumor control and overall survival in melanoma patients. Leukoderma was the primary IL-2-related irAE together with thyroid dysfunctions (70% of irAEs).

Nowadays, immunotherapy with check-point inhibitors has positively augmented the therapeutic opportunities for metastatic melanoma [21], prolonging patient overall survival [22]. However, responses to these therapies are still suboptimal; around 20% for cytotoxic T-lymphocyte antigen-4 (CTLA-4) and 30–40% for PD-1 or PD-L1-targeting monoclonal antibodies, reaching 60% when the two therapies are combined [23,24]. Autoimmune skin reactions are common irAEs of treatment with check-point inhibitors, and development of leukoderma has been associated with a good outcome due to response to therapy [25,26].

By targeting the immune system, immunotherapies induce an indirect effect on tumors that can take time and lead to a delay in the onset of clinical benefits. Therefore, patients usually receive immunotherapy drugs for long periods and beyond conventional progression, unless life-threatening toxicity occurs. Considering this difficulty, a great effort is presently being done to identify markers of response to immunotherapies, both to select patients that would beneficially respond to treatment and to early evaluate patient responsiveness, thus sparing toxicity and resources.

In this review, several aspects of the melanoma/vitiligo relationship are investigated, underlining characteristics of the immune system responses shared by melanoma and vitiligo patients and the value of melanoma-associated leukoderma as a favorable prognostic factor in check-point inhibitor immunotherapy.

## 2. Vitiligo and Spontaneous Melanoma-Associated Leukoderma

Vitiligo is a skin disorder affecting 2% of the world’s population. Progressive destruction of skin melanocytes results into the appearance of patchy depigmentation [13]. Vitiligo pathogenesis is likely autoimmune. Circulating skin-homing melanocyte-specific CD8+ T-lymphocytes and infiltrates of CTLs at the margin of active lesions have been observed in most patients [27]. Both lymphocyte T helper (Th)1 and Th17 responses have been reported [28].

A remarkable aspect of vitiligo is its relationship with cutaneous melanoma. Melanoma-associated leukoderma spontaneously occurs in a fraction of melanoma patients and correlates with a favorable prognosis [26]. A retrospective study indicated that melanoma patients with concomitant leukoderma had a higher survival rate [29]. In some cases, leukoderma appearance revealed a regressing melanoma [30], supporting the necessity of a close examination of patients with skin depigmentation for the presence of primary tumors. Two reports of melanoma arising within a new depigmented patch have been recently published [31].

Some evidence indicates that melanoma-associated leukoderma has clinical features distinct from vitiligo, including advanced age of onset, absence of family history of vitiligo or atopy, equal distribution among men and women, localization of depigmentation to photo exposed areas, and multiple flecked depigmented macules [32,33]. Nevertheless, histological and immunohistological differences have not been found [34,35].

Association between vitiligo and melanoma is thought to be the consequence of an immune response against antigens shared by melanoma and normal melanocytes. Indeed, humoral responses to similar antigens have been proven. In 1995, Cui and Bystryn showed the presence of autoantibodies to melanocytes in 80% of melanoma and in 83% of vitiligo patients. These antibodies were directed to analogous antigens with comparable frequency in both diseases [36]. Moreover, Fishman et al. showed that autoantibodies isolated from vitiligo patients had a destructive effect on melanoma cells both in vitro and in vivo [37]. Other authors reported the presence of autoantibodies against melanocyte differentiation antigens, such as tyrosinase, in the sera of both vitiligo and melanoma patients [38]. As autoantibodies rarely succeed in tumor clearance, the authors proposed that differences in the number of antibodies recognizing these epitopes could characterize vitiligo versus melanoma immune responses, with higher titers in vitiligo patient sera [38].

T cell antigen receptor (TCR) sequencing provides information about T cell antigen specificity because T cell clones having an identical TCR sequence recognize the same antigen. A previous study has shown in vivo accumulation of an identical T cell clone in a primary melanoma and vitiligo-like halo around the tumor [39], supporting the idea that tumor T cells recognizing antigens common to both melanoma and melanocytes may contribute to tumor destruction. Because these CTLs almost never achieve melanoma eradication, similarly to what proposed for autoantibodies, Palermo and colleagues indicated that qualitative differences in CTL reactivity against melanocytes could differentiate vitiligo and cutaneous melanoma, with vitiligo CTLs having a higher affinity to melanocytes [40].

As shown in Figure 1, CTL infiltrate and other immune cell subtypes are similarly represented in melanoma and vitiligo. No data have been reported so far about a direct quantitative comparison of the different immune cell subsets present around the melanoma or the vitiligo lesion. However, vitiligo is characterized by an increased CD8+/CD4+ T lymphocyte ratio. Differently from melanoma, regulatory T cells are decreased in vitiligo or impaired in their functions [41], and presence of myeloid-derived suppressor cells has not been reported. These immunological features underline the existence of an immunosuppressive microenvironment around the melanoma lesion that is not present at the margin of vitiligo and that must be overcome in the onset of melanoma-associated leukoderma.

The MT/ret transgenic mouse model of cutaneous melanoma was used to analyze spontaneous anti-tumor T cell responses [42]. Interestingly, a great number of these mice developed melanoma-associated leukoderma and a correlation was observed between depigmentation development and control of melanoma progression. Mice developing melanoma-associated leukoderma had a higher number of IFN-γ-secreting T cells than mice without skin depigmentation, supporting a crucial role of IFN-γ in the achievement of an effective response against melanoma. More recently, Blenman et al. used the syngeneic mouse model of YUMMER1.7 cell line implanted into C57BL/6J mice [43] and evaluated spontaneous melanoma regression. They found that B cells and neutrophils have a key role in this process [44] and did not report any case of depigmentation.

Using the Wsh mouse model of melanocyte deficiency, Byrne et al. investigated the role of melanoma-associated leukoderma and consequent melanocyte destruction in the maintenance of CTL responses to melanoma. They found that the absence of melanocytes impaired the development of T cell memory towards melanoma, demonstrating that melanocyte antigens liberated by leukoderma drive the long-term functional memory T cell response against melanoma [45].

## 3. Sutton’s Nevus and Halo Phenomenon in Melanoma Patients

Sutton’s nevus (halo nevus) is a nevus surrounded by a symmetric rim of depigmentation, resulting into spontaneous regression of the nevus. Clinical involution of a Sutton’s nevus starts with appearance of a depigmented halo. Thereafter, the central nevus becomes paler and erythematous. In some cases, depigmentation proceeds up to complete nevus regression, leaving only a residual depigmented area. Later, spontaneous re-pigmentation can also occur [46]. Sutton’s nevi are more frequent in childhood and adolescence and could indicate the presence of an active immune system that, eliminating normal and neoplastic melanocytes, prevents tumor development. Vitiligo patients have an increased frequency of halo nevi and multiple halo nevi might predispose to vitiligo onset [47], probably because vitiligo and Sutton’s nevi share a similar autoimmune pathogenetic mechanism [48].

The presence of one or multiple halo phenomena has been occasionally documented in melanoma patients and can occur around melanoma or nevi [49]. However, enough data are not available to estimate the significance of this occurrence. In some cases, halo phenomenon appeared after surgical removal of the primary melanoma lesion [50] or has been reported around cutaneous metastases and scars in melanoma patients with melanoma-associated leukoderma [51].

In Sutton’s nevus, nevi cells are arranged in nests, whereas in melanoma atypical melanocytes are isolated in the epidermis and aggregated in the dermis. In addition, Sutton’s nevus is characterized by cells with rare mitosis and with a consistent lymphocyte infiltration within the nevus, whereas regressing melanoma shows numerous mitotic immature cells and inflammatory infiltrate concentrated at the periphery. In contrast to spontaneous regression of melanoma, halo phenomenon is not associated with fibrosis. This difference could be due to the cytokine microenvironment of the tumor compared to the nevus [52,53,54].

## 4. Melanoma-Associated Leukoderma as an Adverse Effect of Immunotherapy

### 4.1. Therapeutic Vaccination

A high immune infiltrate within the tumor and at the tumor invasive margin, especially when being composed of Th1 cells expressing IFN-γ, as well as CTLs producing granzymes and granulysin, is a common characteristic of tumors having a favorable prognosis. This is true also for melanoma [55]. However, therapeutic attempts to induce melanoma antigen-specific T cells have led to minimal anticancer immune responses. Furthermore, induction of tumor-reactive CTLs was not always sufficient in achieving clinical regression of melanoma [56,57]. Therapeutic vaccination must face different problems; first, the identification of the appropriate antigen to include in the vaccine. Regression of metastatic melanoma with the concomitant occurrence of melanoma-associated leukoderma has been observed so far only in response to immunization with a Melan-A/MART-1 peptide [17]. Vaccination with melanoma-pulsed dendritic cells [58] or in vitro stimulation of patient’s immune cells followed by their re-infusion [19] enhance naturally occurring melanoma antigen-specific immunity and improve clinical results [59]. Interestingly, a patient with metastatic melanoma treated with Melan-A-specific CD8+ lymphocytes developed partial depigmentation associated with skin localization of the infused T cell clones [18]. More recently, clinical trials have been conducted using autologous dendritic cells loaded with autologous tumor antigens that have led to encouraging survival data [60]. No cases of melanoma-associated leukoderma were reported in these trials, supporting the concept that the mechanism of action of these dendritic cell vaccines was the induction of new immune responses to autologous tumor antigens rather than enhancement of existing weak immune responses.

In animal models, immunization against melanoma [61] as well as treatment with a monoclonal antibody against tyrosinase-related protein-1 (TRP-1/gp75) [62] can cause melanoma-associated leukoderma. Development of autoimmune leukoderma was observed in mice treated with anti-CD4 to deplete regulatory T cells (T_reg_) followed by surgery to excise large B16 melanomas [63]. This study underlined the importance of T_reg_ in the prevention of skin autoimmunity in melanoma-bearing hosts.

### 4.2. Check-Point Inhibitors

Leukoderma is an adverse effect of melanoma immunotherapy with check-point inhibitors (Figure 2), with a reported incidence ranging from 3.4% to 28% and a mean onset delay of 30 weeks after therapy initiation [64,65,66,67]. However, even if skin depigmentation is classified as a grade 2 irAE, its appearance does not require therapy discontinuation and could not always be reported.

Notably, depigmentation is significantly associated with a favorable prognosis [25,66,68]. Immunotherapy-induced halo phenomena have been also reported, even if less frequently than leukoderma [69,70]. Appearance of halo nevi in addition to leukoderma might correspond to a stronger anti-melanocyte immune reaction associated with a good prognosis, but the number of reported cases is still too low to permit any conclusive remark. A rare case of regression of benign melanocytic nevi without halo phenomenon was also reported after melanoma therapy with check-point inhibitors [71].

Besides leukoderma, hair depigmentation was also observed [72,73,74]. In the past, patients who underwent immunotherapy for metastatic cutaneous melanoma with adoptive cell transfer of tumor reactive CTLs developed uveitis with diffuse retinal pigment epithelium hypopigmentation resembling Vogt–Koyanagi–Harada syndrome [75]. Thus far, similar side-effects have rarely been reported for immunotherapy of cutaneous melanoma with check-point inhibitors [76].

Development of immunotherapy-related leukoderma does not always associate with the presence of memory T cells that can guarantee a long-term tumor clearance. In one melanoma patient, a complete response to immunotherapy was achieved with the contemporaneous development of leukoderma. After 8 months from therapy conclusion, repigmentation occurred with the appearance of brain and liver metastases [77]. However, in an animal model of melanoma-associated leukoderma [45,63], generation of skin-resident memory T cell responses to melanoma naturally occurred as a result of leukoderma development that played a key role in perpetuating anti-tumor immunity [78].

Leukoderma occurring in melanoma patients treated with check-point inhibitors has some clinical and biological differences with respect to vitiligo. Similarly to what has been previously reported for spontaneously occurring melanoma-associated leukoderma, no family history of vitiligo, thyroiditis, or other autoimmune disorders is reported and the Koebner phenomenon, that is, development of lesions at sites of specifically traumatized skin, seems to be absent [79]. Moreover, higher serum levels of the chemokine CXCL10 are present in melanoma patients developing leukoderma after immunotherapy compared with vitiligo patients or healthy controls [79]. However, as high amounts of CXCL10 characterize vitiligo in an active phase [80], there is the possibility that the reported differences among immunotherapy-induced leukoderma and vitiligo could be less significant when only vitiligo patients in an active phase of the disease are included.

Considering the previously reported data, pre-existence in a patient of an autoimmune disease could be a possible obstacle in the prescription of check-point inhibitor immunotherapy. However, new data suggest that benefits from immunotherapy treatments may outweigh the exacerbation of pre-existing autoimmune disease, especially when the pathology taken into account is not a life-threatening disease such as leukoderma [81].

## 5. Vitiligo and Biomarkers of Response to Immunotherapy with Checkpoint Inhibitors

With only a small subset of melanoma patients responding to immunotherapy with check-point inhibitors, predictive and prognostic biomarkers are urgently needed. Multiple immune features have been evaluated in this search for biomarkers [82,83,84]. Many studies showed an association between a high number of circulating neutrophils and/or neutrophil-to-lymphocyte ratio (NLR) and responsiveness to immunotherapy [85]. Thus, neutrophils may be the expression of an immunosuppressive environment induced by the melanoma itself and their presence could distinguish between responsive and not-responsive individuals. NLR has also been investigated in vitiligo patients. NLR values were found to be significantly higher in patients who had generalized vitiligo than in those with localized vitiligo and healthy controls [86] (Table 1).

Other biomarkers, proposed to be predictive for response to immunotherapy with check-point inhibitors [84], such as PD-1/PD-L1 or CTLA-4 expression, presence of an IFN-γ signature, or augmented inflammatory cytokines, are also hallmarks of active vitiligo (Table 1).

Expression of PD-L1 on tumor cells may play an important role in blocking T cell immune responses. In a study on melanoma patients receiving anti-PD-1 antibodies, intratumoral positivity to PD-L1 significantly correlates with response to immunotherapy [89]. Other evidence indicates that response is associated more with PD-L1 expression in tumor-infiltrating immune cells than on tumor cells themselves [88]. A study of patients with metastatic melanoma showed that exosomes released from melanoma cells carry PD-L1 on their surface, and that the increase in levels of circulating exosomal PD-L1 correlates with tumor response to anti-PD-1 therapy [87]. In vitiligo, PD-1 expression in CD8+ T cells is positively associated with disease activity [90].

An immune-active microenvironment favors the response to check-point inhibitors. High pre-treatment expression of IFN-γ [105] or IFN-γ-inducible factors, such as CXCL9, CXCL10, or CXCL-11, was associated with response in melanoma patients [91]. Interestingly, in vitiligo an IFN-γ signature is present and high serum levels of CXCL-9 [106] or, more prominently, of CXCL-10 are present in patients with progressive disease [92].

IFN-γ uses the Janus kinase (JAK)/signal transducers and activators of transcription (STAT) pathway to activate inflammatory chemokines and cytokines, and expression of both JAK1 and STAT3 is up-regulated in vitiligo [94]. Thus, JAK inhibitors are being evaluating as possible therapeutic options for vitiligo as they down-regulate IFN-γ signaling [107]. Importantly, JAK1 or JAK2 mutations are also associated with acquired resistance to check-point inhibitor immunotherapy in melanoma patients [93].

High pretreatment expression of CTLA-4 in the tumor tissue [88] or in tumor-infiltrating lymphocytes [95] positively correlates with response to treatment with anti-PD-L1 antibodies. Variants in the gene coding for CTLA-4 associate with response to immunotherapy with check-point inhibitor in melanoma patients [96]. The inflammatory response in vitiligo is also thought to be mediated by polymorphism in the *CTLA* gene [97].

The mismatch repair (MMR) system is deputed to the repair of base mismatches occurring during DNA replication [108]. Loss of MMR function leads to microsatellite instability, accumulation of mutations, and production of neoantigens [109]. Moreover, MMR deficiency predicts response to immunotherapy with check-point inhibitors in different tumor types [98,110]. However, no data have been reported so far for melanoma. MMR deficiency has also been linked to vitiligo development. A clinical report indicated that bi-allelic mutations in MMR genes associated with early onset of colorectal cancer also led to vitiligo development [100]. Similarly, a gene associated with vitiligo and identified by differential display between normal and vitiligo patient-derived melanocytes, the “VIT1” gene, is involved in the regulation of MMR functions [99].

An emerging class of biomarkers are microRNAs (miRNAs), which are released from tumor cells into blood circulation. Several tumor-derived miRNAs were found to induce myeloid suppressor cells and predict melanoma patient resistance to immunotherapy with check-point inhibitors and poor survival (miR-146a, miR-155, miR-125b, miR-100, let-7e, miR-125a, miR-146b, miR-99b) [101]. Interestingly, of the miRNAs reported by Huber et al. on melanoma, miR-155 and miR-125b are up-regulated in vitiligo patients with respect to healthy individuals [102]. In addition, let-7e was found to be up-regulated in lesional compared with non-lesional epidermis [104], and miR-146a was up-regulated in the serum of vitiligo mice and vitiligo patients with respect to normal controls [111]. This last miR-146a is over-expressed also in other skin diseases such as in atopic dermatitis, and regulates differentiation of immune cells [112], whereas miR-155 and miR-125b have a role in melanogenesis [102]. Therefore, it is difficult to correlate a patient-positive response to melanoma immunotherapy and the development of immunotherapy-associated leukoderma in the same patient when considering as response indicators only a similar over-expression of specific miRNAs.

## 6. Uveal Melanoma

Uveal melanoma arises from the iris, ciliary body, or choroid of the eye, and represents 3–5% of all melanomas [113,114]. It is the most common primary intraocular malignant tumor in adults, but it is a rare tumor with an incidence of about two to five cases per million. Unfortunately, up to 50% of patients develop metastatic disease, typically in the liver. Uveal melanoma is genetically distinct from cutaneous melanoma, having activating mutations in the *GNAQ* or *GNA11* genes in 80–90% of cases. On the other hand, mutations in *BRAF*, *NRAS,* and *TERT* promoter that are common in cutaneous melanoma are quite absent in uveal tumors. Likewise, monosomy 3 is observed in around 50% of uveal melanomas, whereas it is rarely reported in cutaneous melanoma (Table 2).

Although check-point inhibitors have demonstrated substantial activity in cutaneous melanoma, their effectiveness in uveal melanoma is limited [115,116]. Considering phase II studies with a consistent number of enrolled patients, a multicenter study on 53 uveal melanoma patients receiving the anti-CTLA-4 antibody reported that no patient experienced partial or complete response [117]. A different multicenter study involved 58 patients affected by uveal melanoma and treated with anti-PD-1 or PD-L1 antibodies. In this trial, no complete response was observed, and the overall response rate was of 3.6% [118]. Interestingly, one case of melanoma-associated leukoderma was reported. A similar trial with both anti-PD-1 and anti-CTLA-4 antibodies recorded a response rate of 4.7% [119], and a recent trial on 20 patients reported a partial response for only two patients [116]. Other studies have been performed using check-point inhibitors in monotherapy or combined therapies, but the percentage of response and overall survival was low [115]. At present, only one case of exceptional response to check-point inhibitor therapy was reported [120]. Importantly, this response was accompanied by the development of severe irAEs among which depigmentation resembling Vogt–Koyanagi–Harada syndrome was present [120]. Possibly, more chances of success would come from the use of immunotherapy in an adjuvant setting [121]. Even if skin represented the organ system mainly affected by treatment-dependent irAE, no case of melanoma-associated leukoderma was reported in these studies [121], further sustaining the difference between uveal and cutaneous melanoma in inducing inflammatory and/or immunological responses.

## 7. Melanoma-Associated Leukoderma Beyond Melanoma

Immunotherapy with check-point inhibitors has greatly ameliorated the clinical management of other cancer types besides melanoma and significantly prolonged the overall survival of patients affected by advanced non-small cell lung cancer (NSCLC); head and neck squamous cell carcinoma; Hodgkin’s lymphoma; and urothelial, liver, renal cell, gastric, and colorectal cancer [122]. Different from what was previously reported [123], recent studies indicate that leukoderma development can also occur in patients treated with check-point inhibitors for non-melanoma cancers. The mean onset delay is 31 weeks, similar to what observed in melanoma patients (Figure 3). In 2017, Zarogoulidis et al. reported a case of leukoderma during anti-PD-1 therapy of lung adenocarcinoma [124], and a second case was reported one year later [125] (Figure 3). Leukoderma lesions were also reported in two patients affected by renal cell carcinoma and in patients with cholangiocarcinoma or squamous cell carcinoma [126,127].

## 8. Additional Autoimmune Skin Diseases in Melanoma Patients Treated with Check-Point Inhibitors

Skin reactions are the most common side-effect of melanoma immunotherapy with check-point inhibitors [74,128]. Patients mainly have rash and pruritus (28–37%), whereas cutaneous autoimmune effects different from melanoma-associated leukoderma are rarer (Figure 3) [123]. Generally, these side-effects are treated with corticosteroids and slowly resolve after immunotherapy cessation.

A case of psoriasis vulgaris has been documented in a male patient after 53 weeks of treatment with anti-PD-1 antibody. He had no history of psoriasis or atopic dermatitis [74]. Similarly, a psoriasiform eruption was reported during immunotherapy with anti-PD-1 antibody in a man affected by oral mucosal melanoma, who also had no family or personal history of psoriasis [129]. A third case of psoriasiform eruption was observed in a melanoma patient during the therapy with anti-PD-1 antibody [130]. In addition, four cases of psoriasis exacerbation during immunotherapy were reported [131,132,133]. The characteristics of immunotherapy-induced psoriasis have been analyzed in a multi-center study of tumors treated with anti-PD-1 antibodies [134]. Of the 21 patients examined, 11 were treated for metastatic melanoma. The main clinical type was plaque psoriasis (95%), but six cases of guttate, four of palmoplantar, and one of pustular palmoplantar psoriasis were also observed. A history of psoriasis was the main risk factor to develop psoriasis and if patients had previous psoriasis the timeline to develop immunotherapy-related psoriasis was shorter [134]. The mean onset delay in patients with no previous history of psoriasis is 11 weeks. Interestingly, psoriasis development was also reported in patients affected by other tumor types. Bonigen et al. described 10 cases of psoriasis developed in patients treated with anti-check-point antibodies for lung cancer [134] and three cases of psoriasis were reported in patients diagnosed with NSCLC [135]. The mean onset delay is similar (12.5 weeks) to what observed in melanoma patients (Figure 3).

During melanoma treatment with anti-PD-1 antibodies, a case of lichen planus pemphigoides [136], a case of dermatitis herpetiformis [137], and four cases of bullous pemphigoid [137,138,139] were encountered (Figure 3).

Besides melanoma, one case of lichen planus pemphigoides [140] and one of bullous pemphigoid [141] have been reported for patients treated with check-point inhibitor immunotherapy for NSCLC.

## 9. Conclusions

Immunotherapy with check-point inhibitors has prompted innovation in the treatment of metastatic melanoma as well as of other tumor types. Unfortunately, a large fraction of patients do not respond to immunotherapy or respond only partially. Resolving the natural and acquired resistance to immunotherapy represents the future challenge in cancer research. In the case of melanoma, the objective is the understanding of the immune biology and of the multiple mechanisms of immunosuppression present in patients. Clarification of these molecular mechanisms could lead to identification of novel biomarkers to be used in patient selection and/or follow-up, or to individuate additional therapeutic targets. In fact, a better understanding of immune responses activated by check-point inhibitors could permit the safe combining of immunotherapy with check-point inhibitors with other immunotherapies such as IL-2 treatments, or with different available therapeutic compounds targeting other molecules of the immunosuppression route.

Epidemiological studies on the correlation between melanoma-associated leukoderma and stable disease or overall patient survival have been mainly done within clinical studies with check-point inhibitors. The results clearly indicate that depigmentation is significantly associated with a favorable prognosis. However, pathogenesis of melanoma-associated leukoderma is incompletely understood, and additional studies are required to clarify this aspect. A better comprehension of the pathological mechanisms that lead to leukoderma development in only a subset of melanoma patients would be extremely important to overcome occurrence of resistance to immunotherapy.

In this prospective, vitiligo could represent a parallel disease model system in which the study of patient immune responses against normal melanocytes is in a more advanced phase.

From the data reported above, it is evident that leukoderma development is mainly related to cutaneous melanoma as an irAE of immunotherapy with check-point inhibitors, since the number of cases reported for other cutaneous autoimmune diseases is similar when comparing, as an example, melanoma with lung cancer. Moreover, development of melanoma-associated leukoderma characterizes cutaneous melanoma and it is extremely rare in uveal melanoma.

Furthermore, drawing attention to similarity and differences in the immune responses in cutaneous melanoma compared to uveal melanoma, melanoma-associated leukoderma and vitiligo could lead to acquisition of important knowledge to be applied to melanoma immunotherapeutic approaches.

Finally, it is important to note that appearance of melanoma-associated leukoderma represents not only a sign of effective immune response to immunotherapy with check-point inhibitors but could be also beneficial for the maintenance of such a response through time with generation of anti-melanoma memory T cells. This aspect has been studied in animal models and necessitates additional investigation to verify its significance for melanoma patients.

## Figures and Tables

**Figure 1 ijms-20-05731-f001:**
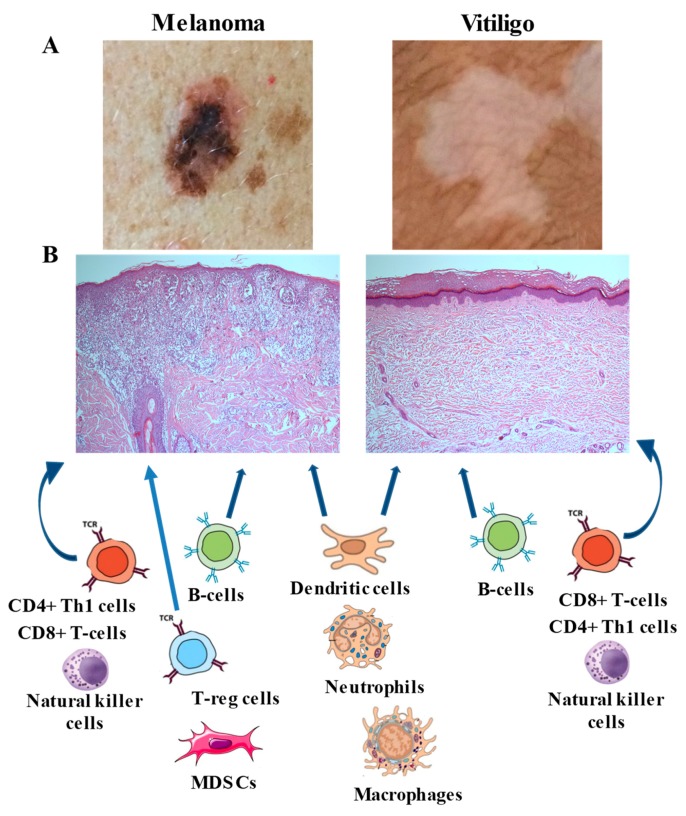
Cytotoxic T lymphocytes (CTLs) present around a melanoma lesion and a vitiligo patch. (**A**) Clinical photographs of a melanoma and a vitiligo lesion. (**B**) Hematoxylin-eosin staining of a skin biopsy section from a melanoma and a vitiligo lesion. These images have been taken for our study approved by IDI-IRCCS Ethical Committee (510/3, 2018), as this figure is from our laboratory and the images have not been published elsewhere. The different subsets of immune cells present around the lesions are schematically represented. Th-1: T helper-1 cells, T-reg: T regulatory cells, MDSCs: myeloid-derived suppressor cells. In vitiligo lesion T-reg cells are reduced and the presence of MDSCs was not reported.

**Figure 2 ijms-20-05731-f002:**
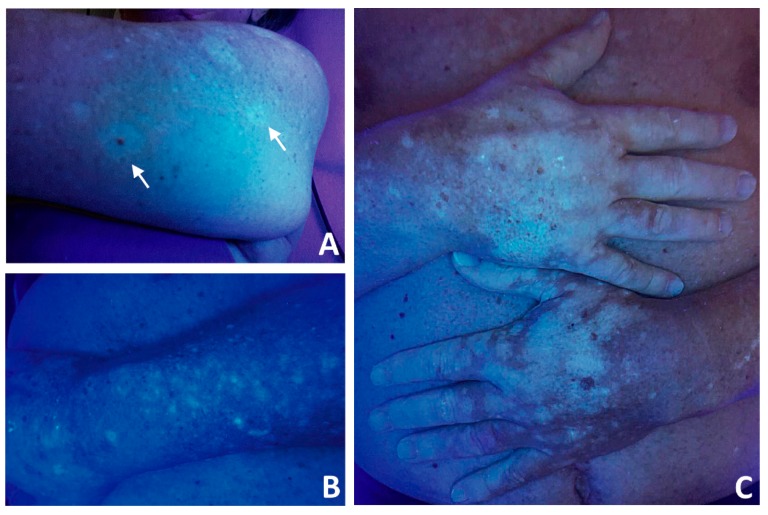
Leukoderma occurring in melanoma patients after treatment with check-point inhibitors. Patients with metastatic melanoma that were treated with check-point inhibitors were enrolled in the study that was approved by the IDI-IRCCS Ethical Committee (510/3, 2018). Photographs have been taken through Wood’s lamp examination. Either halo phenomenon around nevi (arrows, (**A**)) or broad skin patches (**B**,**C**) can be observed. Leukoderma images from two representative patients are shown.

**Figure 3 ijms-20-05731-f003:**
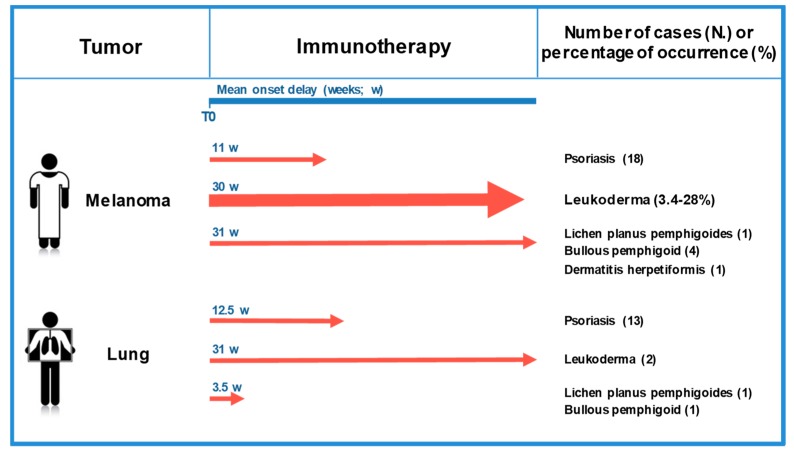
Skin irAEs in patients diagnosed with cutaneous melanoma or lung cancer undergoing immunotherapy with check-point inhibitors. The mean onset delay from therapy initiation (T0) is reported. Leukoderma cases are significantly higher in patients treated with check-point inhibitors for melanoma than for those with lung cancer (thicker arrow). Instead, other autoimmune cutaneous irAEs are similarly represented.

**Table 1 ijms-20-05731-t001:** Biomarkers of response to check-point inhibitor immunotherapy that are also associated with vitiligo development.

Biomarker	Immunotherapy	Vitiligo
NLR	High NLR positively associates with response [85].	High NLR in patients with generalized disease [86].
PD-1/PD-L1	Expression of PD-L1 positively correlates with response [87,88,89].	High levels of PD-1 on CD8+ T cells positively associate with disease activity [90].
IFN-γ and IFN-related genes	Expression of CXCL-9, CXCL-10, CXCL-11 in the tumor microenvironment positively correlates with response [91].	High serum levels of CXCL-9 and CXCL-10 indicate vitiligo active phase [92].
Janus kinase (JAK)/signal transducers and activators of transcription (STAT)	JAK mutations are related to resistance to immunotherapy [93].	JAKs and STATs are over-expressed in vitiligo [94].
CTLA-4	High pretreatment expression of CTLA-4 in tumor tissue [88] or in tumor-infiltrating lymphocytes [95] positively correlates with response. Polymorphisms in the *CTLA-4* gene are associated with response [96].	Polymorphisms in *CTLA-4* gene are involved in vitiligo development [97].
Mismatch repair (MMR)	MMR deficiency positively correlates with response [98].	Vitiligo has been documented in patients with MMR defects [99,100].
microRNAs (miRNAs)	miR-146a, miR-155, miR-125b, miR-100, miR-let-7e, miR-125a, miR-146b, and miR-99b up-regulation predicts resistance to immunotherapy [101].	miR-155, miR-125b, and miR-let-7e are up-regulated in vitiligo [102,103,104].

**Table 2 ijms-20-05731-t002:** Main differences between uveal and cutaneous melanoma and their response to immunotherapy with check-point inhibitors.

UVEAL	CUTANEOUS
606 cases/year in Europe	100,000 cases/year in Europe
Mainly liver metastasis	Metastasis in various organs
*GNAQ* or *GNA11* gene mutations	*BRAF* or *NRAS* gene mutations
Monosomy 3 in 50% of tumors	Monosomy 3 rarely occurring
0.8–5% positive responsiveness to immunotherapy	20–60% positive responsiveness to immunotherapy

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
