# Peer review of "Melanoma and Vitiligo: In Good Company"

_ijms, 2019, doi:10.3390/ijms20225731_

Round 1

Reviewer 1 Report

The manuscript entitled "Melanoma and vitiligo: in good company" is an interesting review about the relationship between melanoma and melanoma-associated leukoderma vitiliginous reaction. This association is well known but it’s pathogenesis is incompletely understood. The most interesting thing would be to evaluate if there is a different behavior in the clinical and biological characteristics and if melanoma-associated leukoderma vitiliginous reaction change prognosis or response to melanoma therapy.

However, the following points may need to be implemented.

In the introduction:

Authors should describe better the relationship between melanoma cells and immune recognition (specify the antigens recognized by CTLS such as tyrosinase, tyrosinase-related protein..), the involved cells mediators (INF..) and the melanoma cells strategies to evade immune recognition (for example the upregulation of the PD1 ligand expression that reduce the T-cell activity..)

In the text:

there is a lack of data about clinical differences between vitiligo and melanoma associated leukoderma reported in the literature. Specify the qualitative differences in CTL reactivity against melanocytes that could differentiate vitiligo and melanoma associated leukoderma.

In the “Sutton’s nevus” section:

Specify that sutton nevus involution might be seen completely only in some case Report the clinical differences between the sutton nevus and melanoma with halo phenomenon referred to the literature.

In “Vitiligo as an adverse effect of melanoma therapy” section:

Report the studies of melanoma-associated leukoderma in patients with advanced disease treated with other therapy such as high-dose interleukin (IL)-2 or interferon-alpha Specify after how long vitiligo like depigmentation appears during melanoma treatment. “Depigmentation is a mild adverse effect not requiring treatment” is not totaly correct. Melanoma-associated leukoderma is classified as Grade 2 according to the guidelines of the Society of Immunotherapy of Cancer and discontinuation of  checkpoint inhibitors is not required but therapy is recommended. It could be interesting to report the possible therapeutic strategies. More than one case of hair (but also eyelashes and eyebrows) depigmentation is reported in literature during checkpoint inhibitors therapy and vitiligo like lesions.

It is necessary a revision work in literature to better define the vitiligo like lesions (and the other cutaneous autoimmune effects) incidence during checkpoint inhibitors.

Author Response

Authors should describe better the relationship between melanoma cells and immune recognition (specify the antigens recognized by CTLS such as tyrosinase, tyrosinase-related protein..), the involved cells mediators (INF..) and the melanoma cells strategies to evade immune recognition (for example the upregulation of the PD1 ligand expression that reduce the T-cell activity..)

Accordingly, data on antigens recognized by CTLS have been added with addition of one reference. In addition, a paragraph on melanoma strategies to evade immune recognition has been added together with three additional references.

In the text:

there is a lack of data about clinical differences between vitiligo and melanoma associated leukoderma reported in the literature. Specify the qualitative differences in CTL reactivity against melanocytes that could differentiate vitiligo and melanoma associated leukoderma.

This aspect has been better underlined in the text also adding a new reference. We now refer to melanoma-associated leukoderma instead to melanoma-associated vitiligo to distinguish the two pathologies.

In the “Sutton’s nevus” section:

Specify that Sutton’s nevus involution might be seen completely only in some case.

The corresponding sentence has been changed.

Report the clinical differences between the Sutton’s nevus and melanoma with halo phenomenon referred to the literature.

This aspect has been better analysed with the addition of references, specifying also in this case when we refer to Sutton’s nevus or to a melanoma-associated halo phenomenon.

In “Vitiligo as an adverse effect of melanoma therapy” section:

Report the studies of melanoma-associated leukoderma in patients with advanced disease treated with other therapy such as high-dose interleukin (IL)-2 or interferon-alpha.

Additional data and a corresponding reference (Curtis B et al. Journal for Immunotherapy of Cancer 2017) have been added in the introduction to clarify this point.

Specify after how long vitiligo like depigmentation appears during melanoma treatment.

This data has been added in the text in addition of the data reported in figure 3.

“Depigmentation is a mild adverse effect not requiring treatment” is not totally correct. Melanoma-associated leukoderma is classified as Grade 2 according to the guidelines of the Society of Immunotherapy of Cancer and discontinuation of checkpoint inhibitors is not required but therapy is recommended. It could be interesting to report the possible therapeutic strategies.

We agree with the Reviewer that depigmentation is a grade 2 adverse effect and we indicated it now in the text. However, in most cases the oncologist chooses both not to discontinuate checkpoint inhibitor therapy and not to give to the patient additional treatments. In addition, experiments on animal models suggest that leukoderma presence, beside indicating the development of an effective immune response against melanoma cells, could also be beneficial to maintain such a response and protect against melanoma relapse (Malik B.T. et al Sci Immunol 2017). A sentence and the corresponding references have been added in the text.

More than one case of hair (but also eyelashes and eyebrows) depigmentation is reported in literature during checkpoint inhibitors therapy and vitiligo like lesions.

A sentence and new references have been added.

It is necessary a revision work in literature to better define the vitiligo like lesions (and the other cutaneous autoimmune effects) incidence during checkpoint inhibitors.

We thought we cited the most up-dated articles on this aspect, but we performed a new research in the literature and added some additional references.

Reviewer 2 Report

Immunotherapy in cancer is of a special significance as a non-invasive, with little if any side-effects and a perspective for a high effectiveness. Melanoma belongs to the most difficult to cure cancers, so new therapeutic approaches to treat it and modifications of existing ones are needed. Therefore, the subject the authors undertook is justified. This is a well-written and scientifically sound manuscript and I have not any serious objection against its publication in IJMS.

Some minor remarks:

Adding of some figures illustrating mutual relationships between melanoma and vitiligo in the context of the disease and therapy would enhance the expression of the manuscript.

The section Conclusion should be rewritten to be more closely related to the subject, e.g. how is vitiligo important in uveal melanoma? In the present form it is too general and says little about the main subject.

The authors should correct section numbering and edit Table 2 caption.

Author Response

Some minor remarks:

Adding of some figures illustrating mutual relationships between melanoma and vitiligo in the context of the disease and therapy would enhance the expression of the manuscript.

Accordingly, a new figure 1 has been added to clarify this aspect.

The section Conclusion should be rewritten to be more closely related to the subject, e.g. how is vitiligo important in uveal melanoma? In the present form it is too general and says little about the main subject.

Conclusion has been changed in order to better focus on the review main subject.

The authors should correct section numbering and edit Table 2 caption.

We corrected both section numbering and figure/table caption.